# Reduced mortality but elevated venous thromboembolism risk following knee and hip arthroplasty in patients with rheumatoid arthritis: A general population-based cohort study

Xinjia Deng[1,2,3], Na Lu[4], Dongxing Xie[1,2,3], Hui Li[1,2,3], Haochen Wang [1,2,3]*

1 Department of Orthopaedics, Xiangya Hospital, Central South University, Changsha, Hunan, China,
2 Key Laboratory of Aging-related Bone and Joint Diseases Prevention and Treatment, Ministry of Education, Xiangya Hospital, Central South University, Changsha, China, 3 Hunan Key Laboratory of Joint Degeneration and Injury, Changsha, Hunan, China, 4 Arthritis Research Canada, Richmond, British Columbia, Canada

* hausenwong@csu.edu.cn

## Abstract

Arthroplasty is indicated for patients with rheumatoid arthritis (RA) who experience significant joint damage, including bone erosions, cartilage degradation and joint deformities. However, studies on its associations with all-cause mortality, cardiovascular disease (CVD), and venous thromboembolism (VTE) among patients with RA are scarce. Our aim was to evaluate the relation of knee arthroplasty or hip arthroplasty to all-cause mortality, relative risk of CVD and incident VTE among patients with RA. We included patients with RA (ages≥20 years) from a large United Kingdom primary care database (i.e., IQVIA Medical Research Database). The primary outcome was all-cause mortality (n = 4,774 for knee arthroplasty, n = 3,362 for hip arthroplasty). The secondary outcomes included incident CVD (n = 4,350 for knee arthroplasty, n = 2,390 for hip arthroplasty) and incident VTE (n = 4,574 for knee arthroplasty, n = 3,174 for hip arthroplasty). We conducted propensity score-matched cohort studies to compare the risks of each outcome between subjects with and without knee arthroplasty (n = 2,387 each) and those with and without hip arthroplasty (n = 1,681 each), respectively. We found that subjects with knee arthroplasty had a 23% lower risk of mortality than those without knee arthroplasty (HR: 0.77, 95%CI: 0.65–0.90). Similarly, a lower, albeit non-statistically significant, risk of mortality was observed among subjects with hip arthroplasty than those without arthroplasty (HR: 0.87, 95%CI: 0.73–1.04). Compared with those without arthroplasty, subjects with knee or hip arthroplasty had a lower risk of CVD. The corresponding HRs were 0.86 (95%CI: 0.73–1.01) and 0.84 (95%CI: 0.69–1.02), respectively. Both subjects with knee or hip arthroplasty showed a higher risk of VTE than their counterparts (HR for knee arthroplasty: 1.63 [95%CI: 1.23–2.17]; HR for hip arthroplasty: 2.19 [95%CI: 1.54–3.11]). The associations of arthroplasty with the risks of mortality, CVD and

**Data availability statement:** All relevant data are within the manuscript and its Supporting information files.

**Funding:** This work was supported by the Fundamental Research Funds for the Central Universities of Central South University (2021zzts0355). The funding source had no role in the design and conduct of the study; collection, management, analysis, and interpretation of the data; preparation, review, or approval of the manuscript; and the decision to submit the manuscript for publication.

**Competing interests:** The authors have declared that no competing interests exist.

VTE were generally consistent across strata of age and sex, with HR ranges from 0.71–3.75 for knee arthroplasty and 0.66–3.36 for hip arthroplasty. In this large population-based cohort of patients with RA, knee arthroplasty was associated with a lower risk of all-cause mortality, while both knee and hip arthroplasty were associated with a higher risk of VTE. No significant associations were observed with CVD. These findings highlight potential long-term benefits and risks of joint replacement in RA, but given the observational design and possibility of residual confounding, the results should be interpreted as associations rather than causal effects. Further studies are warranted to confirm these observations and to better understand the mechanisms underlying these associations.

## Introduction

Rheumatoid arthritis (RA) is one of the most common systemic inflammatory diseases with a prevalence around 0.46% [1]. It is associated with increased risks of cardiovascular disease (CVD, 48%, pooled risk ratio = 1.48 (95% CI 1.36 to 1.62)) [2,3], venous thromboembolism (VTE, 100%, adjusted hazard ratio = 2.0 (95% CI, 1.9–2.2)) [4,5], and mortality (43%, HR = 1.43 (95% CI 1.28 to 1.59)) [6,7]. Epidemiological data indicate that approximately 9% patients with RA are affected by CVD, corresponding with an incidence rate of 3.30 per 100 patient-years (95% CI 2.08–4.25) [8], while VTE occurs around 7.2% of these patients [9]. Despite the advent of biologic therapies and early management strategies, which have substantially improved the prognosis of RA, a significant proportion of patients with RA still experience joint destruction, functional impairment, and pain, necessitating knee or hip arthroplasty [10]. As a result, the rates of knee and hip arthroplasty remain high in this population [11].

The primary objective of knee and hip arthroplasty in patients with RA is to address the structural joint abnormalities, leading to significant improvements in pain and function [12,13]. However, elevated risks of infections and complications like prosthetic loosening persist due to chronic inflammation and immunosuppressive therapies [11]. Based on the anatomic location of their RA, patients with RA may undergo various surgical options, such as tenosynovectomy, radiosynovectomy, arthroscopic surgery, osteotomy, joint fusion, procedure for soft-tissue release, small-joint implant arthroplasty, metatarsal-head excision arthroplasty and total joint replacement [14]. However, given multiple joint involvement, systemic comorbidities (including CVD, VTE, infections, postoperative joint dislocation, and osteoporosis) and polypharmacy, the benefits of these surgeries in terms of quality of life are not always clear among patients with RA [12,15–17]. All-cause mortality is one of the most important indicators for the net risk-benefit effect of any clinical treatment regimens [18]. Previous studies have reported that the rate of mortality after hip and knee arthroplasty surgery is up to 4.8%, with major causes including pulmonary embolism, myocardial infarction, stroke, surgical site infections, prosthetic joint infections, and complications related to anesthesia and bleeding [19–22]. However, for patients with RA, joint

arthroplasty may significantly improve physical function, reduce pain, and enhance overall quality of life. By alleviating disability and decreasing long-term systemic inflammation, such interventions could potentially contribute to a reduction in all-cause mortality among this population over time [23,24]. Therefore, the association of either knee or hip arthroplasty with all-cause mortality in patients with RA specifically remains understudied. Although physical inactivity and use of non-steroidal anti-inflammatory drugs (NSAIDs) are known to be major risk factors for CVD [25], joint arthroplasty may have cardioprotective effects by improving mobility and reducing reliance on analgesics [26,27]. Nevertheless, the impact of these surgeries on CVD risk among patients with RA also needs to be confirmed. In addition, previous studies have reported that knee and hip arthroplasty are associated with an elevated risk of VTE [28]. However, no study has compared VTE incidence in patients with RA undergoing joint arthroplasty with those without joint arthroplasty. The increased susceptibility to VTE in RA is thought to result from chronic systemic inflammation, endothelial dysfunction, reduced mobility, and the use of glucocorticoids or NSAIDs [29–31]. Moreover, the common risk factors for CVD and VTE in patients with RA—including age, gender, body mass index (BMI), RA duration, lifestyle factors, and medications [32–34]—highlight the need for a more thorough understanding of how joint replacement surgery may impact these complications.

The aim of this investigation was to evaluate the relationship of knee arthroplasty or hip arthroplasty to all-cause mortality, risk of CVD and incident VTE among patients with RA. To address these knowledge gaps, we conducted propensity-score matched cohort studies among patients with RA to investigate the relationship of knee or hip arthroplasty with risks of all-cause mortality, CVD and VTE, respectively, while controlling for potential confounders (including age, sex, BMI, lifestyle factors, and other comorbidities). Additionally, we performed subgroup analyses (differential risks by age and sex) to explore potential residual confounding by indication. Based on these, we hypothesized that knee and hip arthroplasty may be associated with lower risks of mortality and CVD but an increased risk of VTE among patients with RA.

## Methods

### Data source

The IQVIA Medical Research Database (IMRD), which incorporates data from The Health Improvement Network (THIN, a Cegedim database), is a UK primary care electronic database that contains anonymised health data on approximately 17 million individuals systematically followed across 558 primary care practices. The information available in IMRD is gathered by general practitioners as part of their routine patient care, subsequently de-identified and incorporated into a central database for research purposes [35]. The Read classification system is employed to code specific diagnoses [36], while drug codes are based on a dictionary derived from the Multilex classification system [37]. An earlier study has demonstrated the validity of the IMRD for use in clinical and epidemiological research studies [38]. This study was reported following the Strengthening the Reporting of Observational studies in Epidemiology (STROBE) Statement. The data were accessed for research purposes on July 14, 2022. We did not have access to information that could identify individual participants during or after data collection. All data has been fully anonymized.

### Study design and cohort assembling

We performed time-stratified, propensity score-matched cohort studies to investigate the association between knee or hip arthroplasty and the risks of all-cause mortality, CVD and VTE. Our study population comprised individuals aged between 20 and 89 years who were diagnosed with RA defined by at least one Read code for RA between January 1995 and December 2018. Read codes for RA have been previously verified in the UK General Practice Research Database, with a positive predictive value (PPV) of around 80%. Approximately 60% of subjects in the UK General Practice Research Database overlap with IMRD [39,40].

We divided the time period between 1995 and 2018 into 24 one-year cohort accrual blocks. Within each cohort accrual block, we identified subjects with incident (new-onset) knee or hip arthroplasty using Read codes and calculated

propensity scores for knee or hip arthroplasty using logistic regression. The variables included in the model were risk factors that were associated with both all-cause mortality and decision making for knee or hip arthroplasty: RA duration, sociodemographic factors (age at time of knee or hip arthroplasty, sex, BMI and socioeconomic status [Townsend Deprivation Index]) [41], lifestyle factors (smoking status and alcohol use), comorbidities (Charlson Comorbidity Index, angina, atrial fibrillation, congestive heart failure, hypertension, ischaemic heart disease, stroke, transient ischaemic attack, myocardial infarction, valvular heart disease, VTE, varicose veins, peripheral vascular disease, other circulatory diseases, dementia, depression, seizure, chronic obstructive pulmonary disease, pneumonia or other infections, diabetes, hyperlipidaemia, osteoporosis, gastritis, gastroesophageal reflux disease, peptic ulcer, liver disease, chronic kidney disease, other inflammatory conditions, fall risk, hip fracture, trauma, and cancer). The covariate assessment period spanned two years prior to index date (defined as the date of knee or hip arthroplasty for surgical subjects and a randomly assigned "pseudo-surgery date" within the same cohort accrual block for non-surgical comparators) for medications and healthcare usage, the most recent visit prior to the index date for sociodemographic and lifestyle factors, and any time before the index date for comorbidities [34,42–45]. This design ensured that both groups were aligned in calendar time and disease course, and that follow-up was measured consistently from a comparable index date rather than from the time of RA diagnosis.

Within each accrual time block, for each participant who underwent knee or hip arthroplasty, we identified a matched comparator without arthroplasty using a greedy matching algorithm [46]. The greedy matching algorithm, implemented through the SAS macro, utilizes a hierarchical digit-based approach to propensity score matching, which inherently defines caliper widths by adjusting precision levels. The specified caliper width is 0.1.

## Assessment of outcomes

The primary outcome of the present study was all-cause mortality, which was determined by the date of death recorded in IMRD, linked to the National Health Service. As soon as a patient's vital status changes to 'dead', the electronic health record is immediately updated and no input is required by the practice staff in IMRD [47]. The secondary outcomes were incident CVD and incident VTE. A patient was considered to have had an incident CVD event at the first recording of any Read term synonymous with myocardial infarction, stroke or heart failure [48,49]. A patient was considered to be a VTE case when he or she had a recorded code of pulmonary embolism or deep vein thrombosis along with either receiving anticoagulant therapy or a fatal outcome within one month of diagnosis [50,51].

## Statistical analysis

On the index date, follow-up began and continued until death, loss to follow-up, or end of the study (December 31, 2018). All-cause mortality and rates of CVD and VTE for each group was calculated by dividing the number of deaths, CVD and VTE by the total person-years of follow-up. The hazard ratio (HR) and related 95% confidence interval (CI) of all-cause mortality and incident CVD and VTE for knee or hip arthroplasty were calculated using Cox proportional hazards regression, respectively. In order to test the proportional hazard assumption, we plotted the cumulative incidence curve for each outcome. If the proportional hazard assumption was violated, we conducted a weighted cox regression to obtain a weighted HR [52].

To explore for potential residual confounding by indication, we examined the relation of knee or hip arthroplasty to the risks of mortality, CVD and VTE according to strata of age (20–69 years, and ≥70 years) and sex using Cox proportional hazards regression. These analyses not only help identify potential effect measure modification, but a difference in mortality related to knee or hip arthroplasty according to age or sex may suggest the presence of potential confounding by unmeasured factors.

All analyses were conducted using SAS V.9.3 (Cary, North Carolina, USA), with two-sided α of 0.05 for significance testing.

## Ethical approval

This study received approval from the medical ethical committee at Xiangya Hospital (2018091077), with waiver of informed consent. All methods were performed in accordance with the relevant guidelines and regulations. This study was approved by the THIN Scientific Review Committee (22SRC023).

## Results

We identified 2,387 matched pairs of subjects with RA (n = 4,774; mean age 66.5 years; 25.2% men; mean BMI 28.1 kg/m$^2$). This group consisted of 2,387 RA patients who had undergone knee arthroplasty and 2,387 matched RA patients without knee arthroplasty. Additionally, we identified 1,681 matched pairs of patients with RA (n = 3,362; mean age 68.2 years; 25.9% men; mean BMI 26.6 kg/m$^2$). This group included 1,681 RA patients who had received hip arthroplasty and 1,681 matched RA patients without hip arthroplasty. All covariates were balanced between the comparison cohorts (S1 Table).

There were 421 deaths (29.9 per 1,000 person-years) in the knee arthroplasty group and 472 deaths (35.8 per 1,000 person-years) in the propensity-score matched group without knee arthroplasty during follow-up. The mean follow-up period was 5.9 years for the knee arthroplasty group and 5.5 years for the non-knee arthroplasty group. Participants with knee arthroplasty had a 23% lower risk of all-cause mortality compared with those without knee arthroplasty (HR = 0.77, 95% CI 0.65 to 0.90; ARR = 1.71%) (Table 1; Fig 1A). We identified 366 deaths (40.6 per 1,000 person-years) in subjects with hip arthroplasty and 395 deaths in those without hip arthroplasty (43.9 per 1,000 person-years) during follow-up. The mean follow-up period was 5.4 years for the hip arthroplasty group and 5.4 years for the non-hip arthroplasty group. The rate of all-cause mortality among subjects with hip arthroplasty was lower, albeit non-statistically significant, than those without hip arthroplasty (HR = 0.87, 95% CI 0.73 to 1.04; ARR = 1.49%) (Table 1; Fig 1B). In general, the associations of knee arthroplasty or hip arthroplasty with the risk of mortality were consistent across the age and sex strata (Table 1).

A total of 183 cases of CVD (myocardial infarction, stroke, or heart failure) occurred in the knee arthroplasty group, and 185 cases of CVD were reported in the comparison group during the follow-up period. Incidence rates per 1,000 person-years for the knee arthroplasty and comparison cohorts were 14.8 and 16.1, respectively. The mean follow-up period was 5.7 years for the knee arthroplasty group and 5.3 years for the non-knee arthroplasty group. Participants with knee arthroplasty had a 14% lower, albeit non-statistically significant, risk of CVD than their counterparts (HR = 0.86, 95% CI 0.73 to 1.01; ARR = 0.09%) (Table 2; Fig 2A). During the follow-up period, 135 cases of CVD in the hip arthroplasty group and 146 CVD cases in the comparison group were identified. Incidence rates per 1,000 person-years for the hip arthroplasty and comparison cohorts were 17.6 and 19.6, respectively. The mean follow-up period was 5.2 years for the hip arthroplasty group and 5.1 years for the non-hip arthroplasty group. The risk of CVD, albeit non-statistically significant, was lower in subjects with hip arthroplasty than those without hip arthroplasty (HR = 0.84, 95% CI 0.69 to 1.02; ARR = 0.77%) (Table 2; Fig 2B). Similar results were observed when the analyses were stratified by age and sex (Table 2).

During the follow-up period, 73 VTE (the combined endpoint of pulmonary embolism and deep vein thrombosis) (5.53 per 1,000 person-years) occurred in the knee arthroplasty group and 50 VTE (3.93 per 1,000 person-years) occurred in the matched group without knee arthroplasty. The mean follow-up period was 5.8 years for the knee arthroplasty group and 5.6 years for the non-knee arthroplasty group. Compared with those without knee arthroplasty, the HR of VTE for subjects with knee arthroplasty was 1.63 (95% CI 1.23 to 2.17) and the corresponding ARR was 0.97% (Table 3; Fig 3A). Similar results were also observed when the association between hip arthroplasty and the risk of VTE was examined (7.98 vs. 3.88 per 1,000 person-years). The mean follow-up period was 5.3 years for the hip arthroplasty group and 5.2 years for the non-hip arthroplasty group. The corresponding HR was 2.19 (95% CI 1.54 to 3.11) and ARR was 2.14% (Table 3; Fig 3B). Results were not significantly modified in the analyses stratified by age and sex (Table 3).

**Table 1. Association between knee and hip arthroplasty with risk of all-cause mortality.**

| | KA | Non-KA | HA | Non-HA |
|---|---|---|---|---|
| Participants, n | 2,387 | 2,387 | 1,681 | 1,681 |
| Events, n | 421 | 472 | 366 | 395 |
| Mean follow-up, y | 5.9 | 5.5 | 5.4 | 5.4 |
| Incidence rate, per 1,000 PY (95%CI) | 29.9 (27.1 to 32.9) | 35.8 (32.7 to 39.2) | 40.6 (36.6 to 45.0) | 43.9 (39.7 to 48.4) |
| HR[a] (95%CI) | **0.77 (0.65 to 0.90)** | 1.00 (reference) | 0.87 (0.73 to 1.04) | 1.00 (reference) |
| **Age < 70 years** | **KA** | **Non-KA** | **HA** | **Non-HA** |
| Participants, n | 1,195 | 1,195 | 675 | 675 |
| Events, n | 132 | 146 | 87 | 95 |
| Mean follow-up, y | 6.5 | 6.1 | 6.0 | 6.0 |
| Incidence rate, per 1,000 PY (95%CI) | 17.1 (14.3 to 20.3) | 20.0 (16.9 to 23.5) | 21.5 (17.3 to 26.6) | 23.6 (19.1 to 28.8) |
| HR[a] (95%CI) | 0.84 (0.66 to 1.06) | 1.00 (reference) | 0.89 (0.67 to 1.21) | 1.00 (reference) |
| **Age ≥ 70 years** | **KA** | **Non-KA** | **HA** | **Non-HA** |
| Participants, n | 689 | 689 | 600 | 600 |
| Events, n | 156 | 183 | 158 | 168 |
| Mean follow-up, y | 4.8 | 4.6 | 4.3 | 4.1 |
| Incidence rate, per 1,000 PY (95%CI) | 47.5 (40.3 to 55.5) | 57.9 (50.0 to 67.0) | 61.8 (52.5 to 72.2) | 68.3 (58.4 to 79.5) |
| HR[a] (95%CI) | **0.82 (0.66 to 1.00)** | 1.00 (reference) | 0.89 (0.73 to 1.12) | 1.00 (reference) |
| **Female** | **KA** | **Non-KA** | **HA** | **Non-HA** |
| Participants, n | 1,657 | 1657 | 1,163 | 1,163 |
| Events, n | 270 | 285 | 238 | 252 |
| Mean follow-up, y | 6.0 | 5.7 | 5.4 | 5.3 |
| Incidence rate, per 1,000 PY (95%CI) | 27.1 (23.9 to 30.5) | 30.4 (27.0 to 34.2) | 37.8 (33.1 to 42.9) | 40.9 (36.0 to 46.3) |
| HR[a] (95%CI) | 0.87 (0.74 to 1.04) | 1.00 (reference) | 0.91 (0.76 to 1.08) | 1.00 (reference) |
| **Male** | **KA** | **Non-KA** | **HA** | **Non-HA** |
| Participants, n | 248 | 248 | 114 | 114 |
| Events, n | 43 | 52 | 22 | 27 |
| Mean follow-up, y | 5.9 | 5.3 | 5.8 | 4.6 |
| Incidence rate, per 1,000 PY (95%CI) | 29.3 (21.2 to 39.4) | 39.7 (29.7 to 52.1) | 33.3 (20.9 to 50.4) | 51.5 (34.0 to 75.0) |
| HR[a] (95%CI) | 0.73 (0.49 to 1.10) | 1.00 (reference) | 0.66 (0.37 to 1.16) | 1.00 (reference) |

KA, knee arthroplasty; HA, hip arthroplasty; PY, person-years; CI, confidence intervals; HR, hazard ratios.

[a]Estimates were time-stratified overlap weighted of propensity score, weighted cox regression using coxphw method were applied if proportional hazard assumption was violated.

## Discussion

In this general practice cohort representative of the UK population, our findings showed a substantially lower risk of all-cause mortality (23% for KA, 13% for HA) in patients with RA after knee or hip arthroplasty, indicating significant protective associations for both KA and HA regarding all-cause mortality. Our study revealed that both KA and HA demonstrated

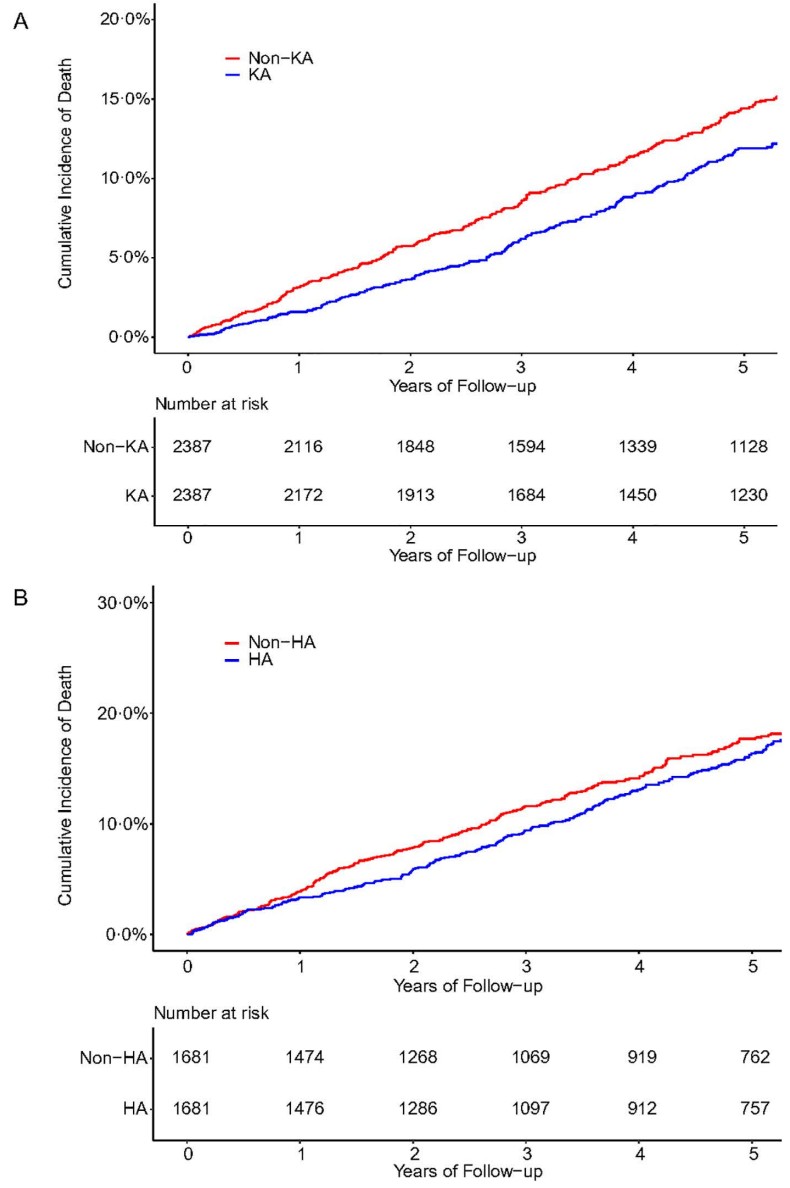

**Fig 1. Cumulative incidence of death among subjects with knee or hip arthroplasty compared to subjects without knee or hip arthroplasty.**
A, subjects with knee arthroplasty compared to subjects without knee arthroplasty; B, subjects with hip arthroplasty compared to subjects without hip arthroplasty. KA, knee arthroplasty; HA, hip arthroplasty.

cardioprotective properties in patients with RA, exhibiting a trend toward reduced cardiovascular disease risk (14% for KA, 16% for HA). However, this potential cardiovascular benefit must be weighed against a substantial elevation in VTE risk, with postoperative analyses showing 63% and 119% increased incidence rates for KA and HA recipients respectively. These results were consistent across age and sex strata. Our findings address a critical gap in the surgical outcomes literature for RA populations and carry important clinical implications for risk-benefit assessments in perioperative decision-making.

**Table 2. Association between knee and hip arthroplasty with risk of cardiovascular disease.**

| | KA | Non-KA | HA | Non-HA |
|---|---|---|---|---|
| Participants, n | 2,175 | 2,175 | 1,465 | 1,465 |
| Events, n | 183 | 185 | 135 | 146 |
| Mean follow-up, y | 5.7 | 5.3 | 5.2 | 5.1 |
| Incidence rate, per 1,000 PY (95%CI) | 14.8 (12.7 to 17.1) | 16.1 (13.8 to 18.6) | 17.6 (14.8 to 20.9) | 19.6 (16.7 to 23.1) |
| HR[a] (95%CI) | 0.86 (0.73 to 1.01) | 1.00 (reference) | 0.84 (0.69 to 1.02) | 1.00 (reference) |
| **Age<70 years** | **KA** | **Non-KA** | **HA** | **Non-HA** |
| Participants, n | 1,140 | 1,140 | 614 | 614 |
| Events, n | 71 | 77 | 41 | 43 |
| Mean follow-up, y | 6.3 | 5.9 | 5.9 | 5.9 |
| Incidence rate, per 1,000 PY (95%CI) | 9.9 (7.8 to 12.5) | 11.6 (9.1 to 14.4) | 11.3 (8.1 to 15.4) | 12.0 (8.7 to 16.1) |
| HR[a] (95%CI) | 0.80 (0.61 to 1.05) | 1.00 (reference) | 0.90 (0.63 to 1.30) | 1.00 (reference) |
| **Age≥70 years** | **KA** | **Non-KA** | **HA** | **Non-HA** |
| Participants, n | 557 | 557 | 451 | 451 |
| Events, n | 64 | 64 | 51 | 51 |
| Mean follow-up, y | 4.4 | 4.3 | 4.1 | 4.0 |
| Incidence rate, per 1,000 PY (95%CI) | 25.9 (19.9 to 33.0) | 27.0 (20.8 to 34.5) | 27.5 (20.5 to 36.1) | 28.5 (21.2 to 37.4) |
| HR[a] (95%CI) | 0.91 (0.69 to 1.19) | 1.00 (reference) | 0.88 (0.62 to 1.21) | 1.00 (reference) |
| **Female** | **KA** | **Non-KA** | **HA** | **Non-HA** |
| Participants, n | 1,566 | 1,566 | 1,058 | 1,058 |
| Events, n | 115 | 120 | 79 | 95 |
| Mean follow-up, y | 5.8 | 5.4 | 5.4 | 5.2 |
| Incidence rate, per 1,000 PY (95%CI) | 12.7 (10.5 to 15.2) | 14.2 (11.7 to 16.9) | 13.8 (10.9 to 17.2) | 17.2 (14.0 to 21.1) |
| HR[a] (95%CI) | 0.88 (0.72 to 1.08) | 1.00 (reference) | **0.76 (0.59 to 0.98)** | 1.00 (reference) |
| **Male** | **KA** | **Non-KA** | **HA** | **Non-HA** |
| Participants, n | 157 | 157 | 47 | 47 |
| Events, n | 16 | 21 | 5 | 6 |
| Mean follow-up, y | 6.2 | 5.3 | 6.0 | 6.1 |
| Incidence rate, per 1,000 PY (95%CI) | 16.6 (9.5 to 26.9) | 25.1 (15.6 to 38.4) | 17.8 (5.8 to 41.4) | 20.9 (7.7 to 45.5) |
| HR[a] (95%CI) | 0.71 (0.40 to 1.28) | 1.00 (reference) | 0.80 (0.31 to 2.04) | 1.00 (reference) |

KA, knee arthroplasty; HA, hip arthroplasty; PY, person-years; CI, confidence intervals; HR, hazard ratios.

[a]Estimates were time-stratified overlap weighted of propensity score, weighted cox regression using coxphw method were applied if proportional hazard assumption was violated.

## Comparison with previous studies

Two studies reported higher mortality among patients with RA who had knee or hip arthroplasty than the general population with [53] or without joint surgery [54], but such comparisons are prone to confounding by indication [55], as RA itself

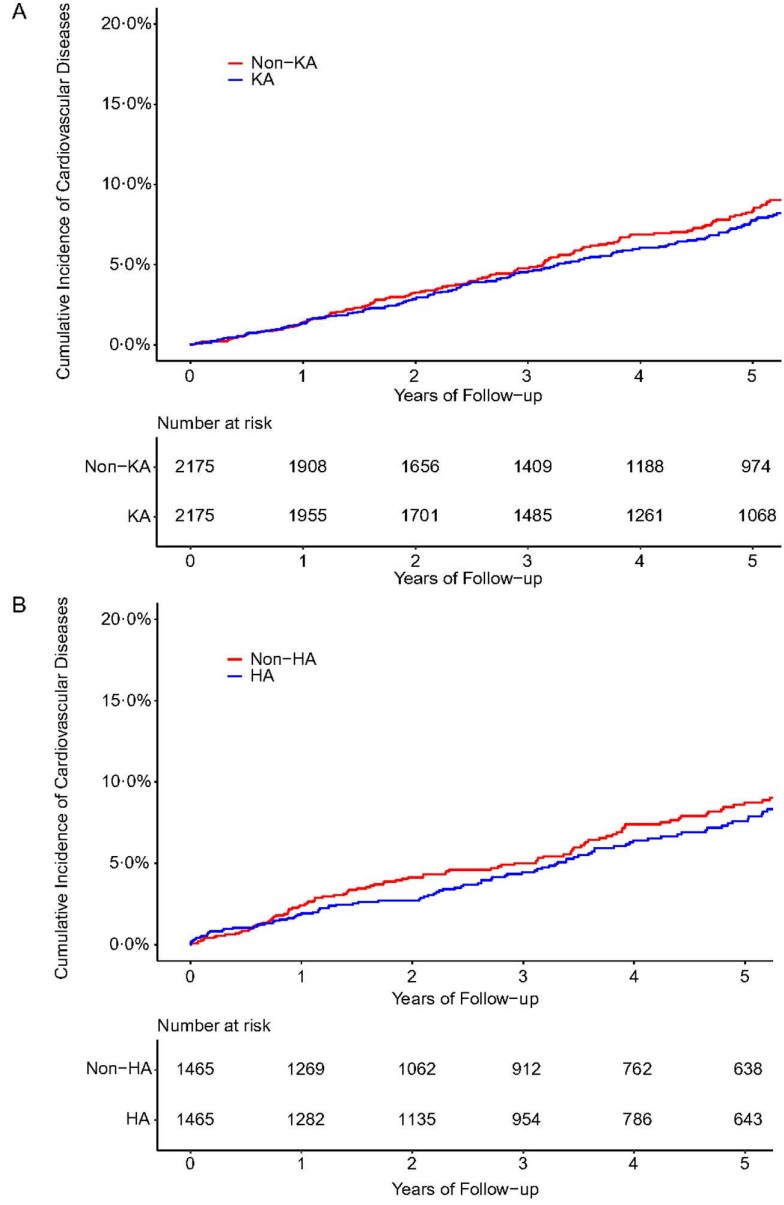

**Fig 2. Cumulative incidence of cardiovascular disease among subjects with knee or hip arthroplasty compared to subjects without knee or hip arthroplasty.** A, subjects with knee arthroplasty compared to subjects without knee arthroplasty; B, subjects with hip arthroplasty compared to subjects without hip arthroplasty. KA, knee arthroplasty; HA, hip arthroplasty.

carries higher baseline mortality [56]. One retrospective study of 424 patients found that there was no survival benefit from the joint surgeries among patients with RA [57]. However, the joint surgeries in this study consisted of heterogeneous surgical procedures, including total joint arthroplasty (51.4%) and other joint reconstructive operation; thus, the study did not specifically assess the survival benefits of knee or hip arthroplasty among subjects with RA. Findings from the Danish Hip Arthroplasty Registry showed overall survival of primary THA in RA patients comparable to that in OA, but did not evaluate survival benefits within RA (registry study) [58]. Another study showed that the risk of myocardial infarction among patients

**Table 3. Association between knee and hip arthroplasty with risk of venous thromboembolism.**

| | KA | Non-KA | HA | Non-HA |
|---|---|---|---|---|
| Participants, n | 2,287 | 2,287 | 1,587 | 1,587 |
| Events, n | 73 | 50 | 67 | 32 |
| Mean follow-up, y | 5.8 | 5.6 | 5.3 | 5.2 |
| Incidence rate, per 1,000 PY (95%CI) | 5.53 (4.34 to 6.96) | 3.93 (2.92 to 5.18) | 7.98 (6.19 to 10.14) | 3.88 (2.65 to 5.48) |
| HR[a] (95%CI) | **1.63 (1.23 to 2.17)** | 1.00 (reference) | **2.19 (1.54 to 3.11)** | 1.00 (reference) |
| **Age <70 years** | **KA** | **Non-KA** | **HA** | **Non-HA** |
| Participants, n | 1,182 | 1,182 | 614 | 614 |
| Events, n | 35 | 25 | 11 | 3 |
| Mean follow-up, y | 6.3 | 6.1 | 6.0 | 5.9 |
| Incidence rate, per 1,000 PY (95%CI) | 4.7 (3.3 to 6.6) | 3.5 (2.2 to 5.1) | 3.0 (1.5 to 5.3) | 0.8 (0.2 to 2.4) |
| HR[a] (95%CI) | 1.41 (0.95 to 2.09) | 1.00 (reference) | 2.33 (0.83 to 6.54) | 1.00 (reference) |
| **Age ≥70 years** | **KA** | **Non-KA** | **HA** | **Non-HA** |
| Participants, n | 649 | 649 | 505 | 505 |
| Events, n | 22 | 16 | 19 | 11 |
| Mean follow-up, y | 4.7 | 4.4 | 4.2 | 4.2 |
| Incidence rate, per 1,000 PY (95%CI) | 7.2 (4.5 to 10.9) | 5.6 (3.2 to 9.1) | 9.0 (5.4 to 14.1) | 5.2 (2.6 to 9.2) |
| HR[a] (95%CI) | 1.43 (0.87 to 2.33) | 1.00 (reference) | 1.78 (0.98 to 3.24) | 1.00 (reference) |
| **Female** | **KA** | **Non-KA** | **HA** | **Non-HA** |
| Participants, n | 1,563 | 1,563 | 1,054 | 1,054 |
| Events, n | 46 | 35 | 51 | 15 |
| Mean follow-up, y | 5.9 | 5.6 | 5.2 | 5.4 |
| Incidence rate, per 1,000 PY (95%CI) | 5.4 (3.9 to 7.1) | 3.8 (2.6 to 5.2) | 9.3 (6.9 to 12.2) | 2.7 (1.5 to 4.4) |
| HR[a] (95%CI) | 1.31 (0.93 to 1.85) | 1.00 (reference) | **3.36 (2.08 to 5.42)** | 1.00 (reference) |
| **Male** | **KA** | **Non-KA** | **HA** | **Non-HA** |
| Participants, n | 269 | 269 | 114 | 114 |
| Events, n | 15 | 4 | 4 | 3 |
| Mean follow-up, y | 5.4 | 5.1 | 5.6 | 5.6 |
| Incidence rate, per 1,000 PY (95%CI) | 10.3 (5.8 to 16.9) | 2.9 (0.8 to 7.4) | 6.3 (1.7 to 16.1) | 4.7 (1.0 to 13.7) |
| HR[a] (95%CI) | **3.75 (1.52 to 9.23)** | 1.00 (reference) | 2.00 (0.56 to 7.09) | 1.00 (reference) |

KA, knee arthroplasty; HA, hip arthroplasty; PY, person-years; CI, confidence intervals; HR, hazard ratios.

[a]Estimates were time-stratified overlap weighted of propensity score, weighted cox regression using coxphw method were applied if proportional hazard assumption was violated.

with RA who had knee or hip arthroplasty was similar to the general population without joint surgery [53]; however, this study could also be suspectable to confounding by indication bias [55]. In the present study, we addressed this issue by restricting our participants to those with RA, excluding subjects who were deemed ineligible for joint arthroplasty due to mortality risk and by using propensity score matching to minimise confounding.

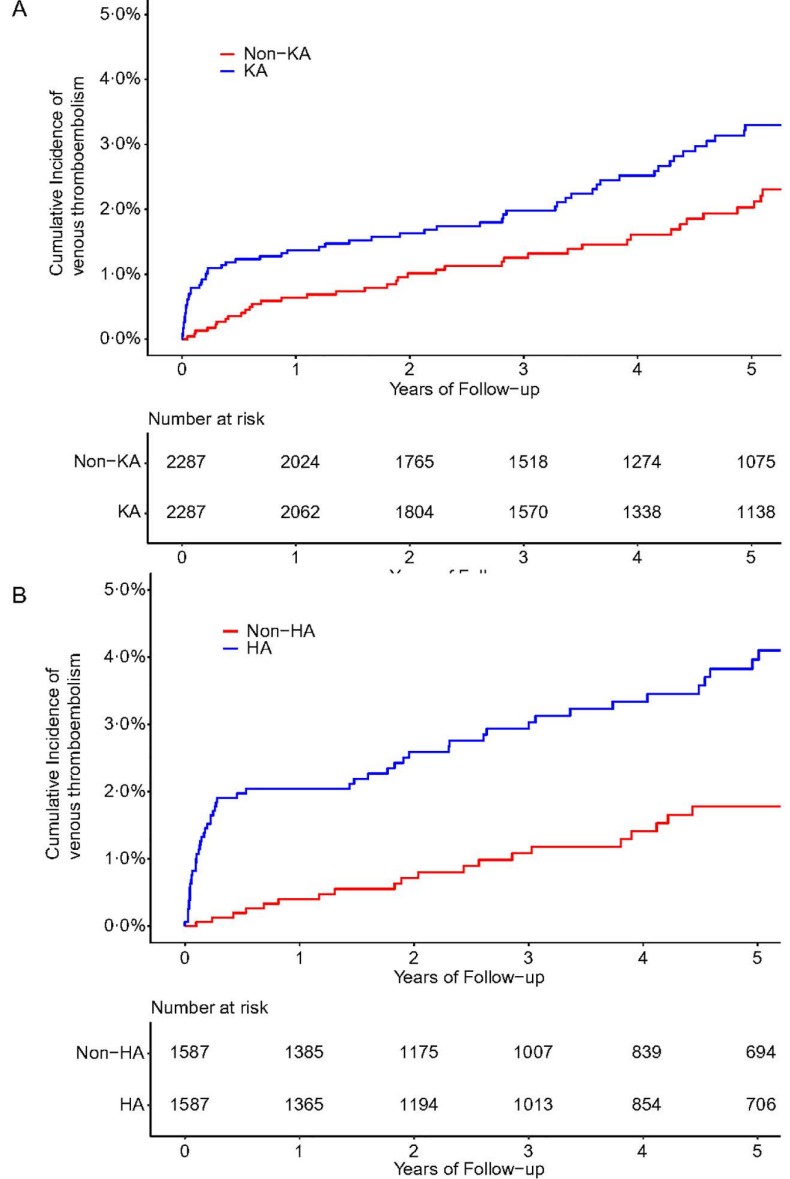

**Fig 3. Cumulative incidence of venous thromboembolism among subjects with knee or hip arthroplasty compared to subjects without knee or hip arthroplasty.** A, subjects with knee arthroplasty compared to subjects without knee arthroplasty; B, subjects with hip arthroplasty compared to subjects without hip arthroplasty. KA, knee arthroplasty; HA, hip arthroplasty.

### Possible explanations

Several mechanisms have been proposed to explain the potential protective effect of knee or hip arthroplasty on all-cause mortality and CVD. First, knee or hip arthroplasty reduces the pain and improves psychological stress in the majority of patients with RA [12]. Since pain and psychological stress are associated with an increased risk of all-cause mortality [59,60] and cardiovascular events [61], knee or hip arthroplasty may lower the risk of these diseases through those mechanisms. Second, knee or hip arthroplasty may improve the physical function and capability for physical activity [13,62]. Increased levels of physical activity have been demonstrated to have survival benefits and cardiovascular protective effect [63–65]. Third,

patients with RA undergoing arthroplasty may inherently have better baseline health or access to healthcare. Fourth, arthroplasty may attenuate systemic inflammation, a key driver of CVD in RA. Chronic inflammation, marked by elevated cytokines (e.g., IL-6, TNF-α) and acute-phase reactants like CRP, accelerates endothelial dysfunction and plaque instability [32,66]. By alleviating localized joint inflammation and reducing inflammatory burden, arthroplasty may lower inflammatory burden, indirectly improving cardiovascular outcomes. As to the higher risk of VTE after knee or hip arthroplasty among patients with RA, venous stasis, hypercoagulability, and endothelial injury from the joint surgery could be the underlying mechanism [28,67].

## Strengths and limitations

This study has several strengths and limitations. First, our results were generated from a population-based sample; thus, the findings are generalizable to other populations with similar characteristics. Second, using incident knee and hip arthroplasty minimises the selection bias that could otherwise underestimate the risk of death if prevalent knee and hip arthroplasty were included in the analysis [68]. Third, similar results were observed when the analyses were stratified by age and sex, which may help to indicate the reliability of this study.

Several limitations of this study are worth noting. First, population-based studies such as those performed in IMRD often lack information on disease activity measures and disease characteristics. Although our study was restricted among subjects with RA, we were unable to assess the stage or gravity of disease between patients with and without knee or hip arthroplasty due to the constraints of the database used in this study. It is possible that patients with more severe disease have increased all-cause mortality and risk of CVD. Furthermore, due to the limitations of the IMRD database, we were unable to include data on the clinical pathways before arthroplasty, radiographs for the assessment of joint injury severity, VTE prophylaxis, imaging for DVT, preoperative cardiac evaluation and anticoagulation controlled, surgical details (e.g., tourniquet time, surgical time, type of implants, cementation or non-cementation and use of elastic stocking), and post-operative complications such as anemia and blood transfusions, which limits the scope of our analysis. Second, lack of sufficient information on the precise cause of death from some subjects precluded us from investigating the association of knee or hip arthroplasty with the risk of the cause-specific death. Nevertheless, the overall all-cause mortality trends are critically important in their own right, as mortality represents the overall net health outcome of various benefits and risks associated with disease management [18]. Third, the current data set does not contain information on the period of biological disease-modifying anti-rheumatic drugs (DMARDs) administration. Since studies have shown that biological DMARDs use was associated a decreased risk of mortality and cardiovascular events in patients with RA [69,70], without adjustment of biological DMARD could also introduce potential residual confounding. Fourthly, we are unable to distinguish between unilateral and bilateral cases, and therefore cannot further explore the distinct impacts of unilateral and bilateral procedures on the outcomes. Fifth, we did not assess short-term postoperative complications (e.g., infections), future studies should be specifically designed to systematically investigate these outcomes.

## Clinical and research implications

Joint arthroplasty is elective yet invasive; prognosis is central to candidacy, and mortality is a key patient-centred outcome [54]. In this first study to show lower mortality after knee or hip arthroplasty in RA, our findings suggest that timely surgery may confer systemic benefits beyond joint restoration. Signals of cardioprotection further support attention to RA-related disability when preventing common comorbidities such as CVD, while the higher VTE risk mandates rigorous perioperative planning and post-operative monitoring. Subgroup results were consistent by age and sex, implying RA-specific systemic factors—persistent inflammation and immune dysregulation—rather than age/sex biology, drive postoperative CVD/VTE risks [30,71,72]. Perioperative contributors (immobilization, surgical trauma, transfusion) [73] and immunosuppressive therapies may further and uniformly modulate these risks [29,31]. These results underscore a careful balance between long-term prognostic gains and acute perioperative hazards and should be fully integrated into shared decision-making to optimize outcomes.

Age and BMI are the critical determinants of surgical prognosis. Although older age is generally linked to higher postoperative complications after arthroplasty, prior studies suggest age variation does not substantially influence the likelihood of undergoing joint replacement in RA, with older patients sometimes receiving surgery earlier after diagnosis [74,75]. Obesity is another well-established risk factor for an increased burden of comorbidities. But a previous study has reported that BMI does not affect the risk of joint arthroplasty in patients with RA, which may be due to obese patients receiving more comprehensive preoperative assessments and more proactive management of complications [76]. Ethnic minorities and individuals from lower socioeconomic backgrounds may experience disparities in access to surgery and postoperative care, potentially affecting outcomes [77–79]. Similarly, malnutrition, common in severe RA, may further worsen complications and mortality [80,81].

Patients with RA are often in poorer physical condition compared to healthy individuals of a similar age. Additionally, many medications used to treat RA are related to immunosuppression, which elevates the mortality associated with surgery [82]. Length of stay and the risk of complications during hospitalization in patients with RA undergoing orthopaedic procedures appears increased [82,83]. In arthroplasty, prolonged hospitalization—driven by immobilization, systemic inflammation, comorbidities, rehabilitation demands, and surgical stress—may raise mortality, CVD, and VTE risks [83,84]. Accordingly, care should prioritize prompt thromboprophylaxis, cardiovascular risk mitigation, and early mobilization, with rigorous perioperative optimization to reduce complications from extended stays [84,85]. Although comprehensive perioperative programs require upfront resources, they may be cost-saving by preventing downstream cardiovascular and thromboembolic events [86,87]. VTE prevention should be multimodal [88]: mechanical measures (intermittent pneumatic compression, graduated stockings) for high-bleeding-risk patients, and pharmacologic agents (LMWH, DOACs) with proven efficacy and acceptable safety [89,90]. Multimodal prophylaxis, guided by risk assessment tools like Caprini or RAPT, optimizes outcomes by balancing thromboprophylaxis benefits against bleeding risks, particularly in high-risk RA with chronic inflammation or immobilization [91]. Coordinated, interdisciplinary management across medicine, surgery, anaesthesia, and nursing is essential to implement these strategies [11,73,92].

## Conclusions

In this large population-based cohort of patients with RA, knee arthroplasty was associated with a lower risk of all-cause mortality, while both knee and hip arthroplasty were associated with a higher risk of VTE. No significant associations were observed with CVD. These findings highlight potential long-term benefits and risks of joint replacement in RA, but given the observational design and possibility of residual confounding, the results should be interpreted as associations rather than causal effects. Further studies are warranted to confirm these observations and to better understand the mechanisms underlying these associations.

## Supporting information

**S1 Table. Baseline characteristics in the propensity score-matched cohort.**
(S1 Table.DOCX)

**S1 File. STROBE_checklist.**
(PDF)

## Author contributions

**Conceptualization:** Xinjia Deng, Haochen Wang.

**Data curation:** Na Lu, Dongxing Xie.

**Formal analysis:** Na Lu, Hui Li.

**Funding acquisition:** Xinjia Deng.

**Investigation:** Xinjia Deng, Haochen Wang.

**Methodology:** Xinjia Deng, Haochen Wang.

**Project administration:** Haochen Wang.

**Supervision:** Haochen Wang.

**Writing – original draft:** Xinjia Deng, Haochen Wang.

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
