## [Decision Letter · Decision Letter 0]

18 Mar 2025

Dear Dr. Wang,

Thank you for submitting your manuscript to PLOS ONE. After careful consideration, we feel that it has merit but does not fully meet PLOS ONE’s publication criteria as it currently stands. Therefore, we invite you to submit a revised version of the manuscript that addresses the points raised during the review process.

The reviewers who assessed your manuscript have suggested changes and expressed concerns, which you can find in the attached document. I kindly ask you to evaluate the suggested changes and respond to the questions raised in order to improve the text and enhance the publishability of your work.

We look forward to receiving your revised manuscript.

Kind regards,

Gennaro Pipino, Md

Academic Editor

PLOS ONE

Journal Requirements:

Reviewers' comments:

Reviewer's Responses to Questions

**Comments to the Author**

1. Is the manuscript technically sound, and do the data support the conclusions?

Reviewer #1: Yes

Reviewer #2: Partly

2. Has the statistical analysis been performed appropriately and rigorously?

Reviewer #1: Yes

Reviewer #2: N/A

3. Have the authors made all data underlying the findings in their manuscript fully available?

Reviewer #1: Yes

Reviewer #2: Yes

4. Is the manuscript presented in an intelligible fashion and written in standard English?

Reviewer #1: Yes

Reviewer #2: Yes

Reviewer #1: The manuscript is a valuable contribution to the field of orthopaedic surgery and rheumatoid arthritis research. The statistical approach is sound, and the conclusions are well-supported by the data. I commend the authors for their strong work.

Addressing the clarifications suggested below may further strengthen the manuscript:

Abstract

The key findings are well summarized, but the risk estimates for CVD are not statistically significant. Consider emphasizing that the trend suggests a reduction in risk but is not conclusive.

The sample size should be reiterated clearly to highlight the study's strength.

Introduction

The background is well-written, but explicitly defining the study's hypothesis earlier would improve readability.

Line 26: “Arthroplasty surgery is indicated for rheumatoid arthritis (RA) patients…” → Consider revising to "Arthroplasty is indicated for patients with rheumatoid arthritis (RA)..." (avoid "RA patients" for better readability).

Consider adding a brief statement acknowledging previous research that found no survival benefit to provide better contrast with the study’s findings.

Methods

Line 104: Clarify how loss to follow-up was handled. Were these patients censored, and if so, at what time point?

Line 133-140: The matching process could be better explained, particularly how greedy matching was implemented (e.g., caliper width used).

Results

Line 197: Consider reporting absolute risk reduction (ARR) along with HRs to aid clinical interpretation.

Tables 1-3: Ensure consistency in presenting statistical significance (bold or asterisk for p < 0.05).

Table 1 footnotes: Ensure uniform formatting for hazard ratios (HR) and confidence intervals (CI).

Discussion

Line 275: “Our findings also showed that either knee or hip arthroplasty decreased the risk of CVD and increased the risk of VTE...” → Consider specifying the magnitude of risk reduction/increase directly in the sentence.

Line 279: The discussion on mechanisms is strong but should also consider alternative explanations (e.g., RA patients undergoing arthroplasty may have better baseline health or access to care).

Line 312: The authors should acknowledge potential unmeasured confounders, such as biologic DMARD use and socioeconomic factors.

Line 374: The statement "coordinated care involving medicine, surgery, anaesthesia, and nursing throughout the perioperative process..." is important. Consider referencing specific perioperative guidelines for RA patients.

Conclusion

The conclusion is appropriate but could briefly emphasize clinical implications (e.g., "These findings suggest that arthroplasty may be considered not only for joint function but also for potential survival and cardiovascular benefits in carefully selected RA patients.").

Overall, well done on a well-conducted study. These minor refinements will further enhance clarity, reproducibility, and clinical relevance.

Reviewer #2: Title

It did not emphasize the highlights of the study

The terms" survival benefits" were non-specific. It can be improved.

Abstract

Row 27- What it means " significant joint damage"?

Row 29- Which type of " risk" was evaluated?

Introduction

Row 56- the "increased risks" could be written with nunmbers and percentages.

It is lacking information about advanvantages and disadvantages related with TKA and THA in RA patients. It can be improved.

The common complications related with TKA and THA in RA were not written, mainly VTE and CVD. It can be improved.

The authors did not mentioned the rate of infection in RA patients. Why not?It could be related or cause of mortality? Nothing was written.

Row 94 and row 97: What it means" all cause mortality"? It is suggested to discriminate the main causes of death, included infection.

Row 95: Which classification system was used to determine the level of RA? Nothing was written.

Row 98: Which were the " potential confounders" factors?

Methods

Row 122 -Why the authors included a broad spectrum for age? The results for TKA and THA between young and older patients are similar?. I suggested to explain in more details.

Row 123-124: The RA codes was the same in the period of study(1995-2018)?Nothing was written.

Row 140,141 and 143: What it means the term " other"? I suggest to describe with more information.

Row 155-158: How the authors correlated the causality among death with joint arthroplasty in RA patients? It remains unclear.

The use and tourniquet, surgical time for TKA, type of implants( constrained or not), cementation or uncementation, use of elastic stocking, VTE drug prophylaxis were not mentioned in this section. Why not?

The grade and stage of disease related with RA patients, were not written. It is suggested to add more information.

Results

The postoperative time after TKA and THA related with primary and secondary outcome were not written in the descriptive manner. Why not?

The correlation between BMI and VTE were not mentioned? It is suggested to add more information.

The use of medications*like corticoids) and period of time related with RA were not mentioned? It is suggested to explain in more details.

The clinical pathways before THA and TKA between 1995 and 2018 were not written. Why not?

Tables 1,2,3

Why the authors separated the variable age above and under 70ys? It is suggested to add references to support.

Discussion

THis section is short. It can be improved

the main findings were written in the first paragraph in the generic manner. It can be improved.

The subgoup analysis and correlation with risk factors could be more explored.

The VTE prophylaxis included mechanical and medications were not written in this section. Why not?

The period of hospitalization and correlation with primary and secondary outcomes remains undetermined,

The clinical relevance could be emphasized.

Conclusion and references OK

**Do you want your identity to be public for this peer review?** For information about this choice, including consent withdrawal, please see our Privacy Policy

Reviewer #1: No

Reviewer #2: No

---

## [Author Response · Author response to Decision Letter 1]

29 Apr 2025

Editors’ Comments to the Authors:

After careful consideration, we feel that it has merit but does not fully meet PLOS ONE’s publication criteria as it currently stands. Therefore, we invite you to submit a revised version of the manuscript that addresses the points raised during the review process.

The reviewers who assessed your manuscript have suggested changes and expressed concerns, which you can find in the attached document. I kindly ask you to evaluate the suggested changes and respond to the questions raised in order to improve the text and enhance the publishability of your work.

Response: We appreciate the editor’s comment. We provide a point-by-point response to the reviewers’ comments. We hope that our responses are satisfactory and that the changes we have made in the text reflect our responsiveness to the comments and suggestions.

Reviewer #1:

Comments to the Author

The manuscript is a valuable contribution to the field of orthopaedic surgery and rheumatoid arthritis research. The statistical approach is sound, and the conclusions are well-supported by the data. I commend the authors for their strong work.

Response: We are very grateful to the reviewer for the valuable comments and suggestions that have helped improve both the content and presentation of our work. Per the reviewer’s suggestions, the manuscript has been reviewed and reorganized.

Comment 1

Abstract

The key findings are well summarized, but the risk estimates for CVD are not statistically significant. Consider emphasizing that the trend suggests a reduction in risk but is not conclusive.

Response: We appreciate the reviewer’s suggestions. Per the reviewer’s suggestion, we have modified the description of findings in the Abstract section to emphasize that the trend suggests a reduction in risk but is not conclusive.

Action: “Our population-based cohort study provides the first evidence that knee and hip arthroplasty are associated with a lower risk of mortality and a trend toward reduced CVD risk but an increased risk of VTE among patients with RA.” (Page 3, line 52-55, in the clean copy of the revised manuscript)

Comment 2 The sample size should be reiterated clearly to highlight the study’s strength.

Response: Many thanks for the reviewer’s suggestion. We have added the description of the sample size on all-cause mortality, cardiovascular disease (CVD), and venous thromboembolism (VTE) in the Abstract section.

Action: “The primary outcome was all-cause mortality (n=4,774 for knee arthroplasty, n=3,362 for hip arthroplasty). The secondary outcomes included incident CVD (n=4,350 for knee arthroplasty, n=2,390 for hip arthroplasty) and incident VTE (n=4,574 for knee arthroplasty, n=3,174 for hip arthroplasty).” (Page 2, line 34-37, in the clean copy of the revised manuscript)

Comment 3

Introduction

The background is well-written, but explicitly defining the study’s hypothesis earlier would improve readability.

Line 26: “Arthroplasty surgery is indicated for rheumatoid arthritis (RA) patients…” →Consider revising to “Arthroplasty is indicated for patients with rheumatoid arthritis (RA)...” (avoid “RA patients” for better readability).

Response: We appreciate the reviewer’s comments. Per the reviewer’s comment, we have revised the wording of this sentence in accordance with your suggestion in the Abstract section of the manuscript.

Action: “Arthroplasty is indicated for patients with rheumatoid arthritis (RA) who experience significant joint damage, including bone erosions, cartilage degradation and joint deformities.” (Page 2, line 27-29, in the clean copy of the revised manuscript)

Comment 4

Consider adding a brief statement acknowledging previous research that found no survival benefit to provide better contrast with the study’s findings.

Response: We appreciate the reviewer’s comment. Previous literature has suggested that patients with RA undergoing arthroplasty may present with comorbidities such as infections and osteoporosis 1,2, while no studies have yet investigated the survival benefits of arthroplasty among patients with RA. We have revised the corresponding content in the Introduction section.

References:

1 Ravi, B. et al. A systematic review and meta-analysis comparing complications following total joint arthroplasty for rheumatoid arthritis versus for osteoarthritis. Arthritis Rheum 64, 3839-3849, doi:10.1002/art.37690 (2012).

2 Xiao, P. L. et al. Prevalence and treatment rate of osteoporosis in patients undergoing total knee and hip arthroplasty: a systematic review and meta-analysis. Arch Osteoporos 17, 16, doi:10.1007/s11657-021-01055-9 (2022).

Action: “However, given multiple joint involvement, systemic comorbidities (including CVD, VTE, infections, postoperative joint dislocation, and osteoporosis) and polypharmacy, the benefits of these surgeries in terms of quality of life are not always clear among patients with RA 12,15-17.” (Page 4, line 82-85, in the clean copy of the revised manuscript)

References:

12 Burn, E. et al. The effect of rheumatoid arthritis on patient-reported outcomes following knee and hip replacement: evidence from routinely collected data. Rheumatology (Oxford) 58, 1016-1024, doi:10.1093/rheumatology/key409 (2019).

15 Momohara, S. et al. Efficacy of total joint arthroplasty in patients with established rheumatoid arthritis: improved longitudinal effects on disease activity but not on health-related quality of life. Mod Rheumatol 21, 476-481, doi:10.1007/s10165-011-0432-9 (2011).

16 Ravi, B. et al. A systematic review and meta-analysis comparing complications following total joint arthroplasty for rheumatoid arthritis versus for osteoarthritis. Arthritis Rheum 64, 3839-3849, doi:10.1002/art.37690 (2012).

17 Xiao, P. L. et al. Prevalence and treatment rate of osteoporosis in patients undergoing total knee and hip arthroplasty: a systematic review and meta-analysis. Arch Osteoporos 17, 16, doi:10.1007/s11657-021-01055-9 (2022).

Comment 5

Methods

Line 104: Clarify how loss to follow-up was handled. Were these patients censored, and if so, at what time point?

Response: We appreciate the reviewer’s suggestion. The loss to follow-up will be treated as censored data, an approach that has been documented in the statistical analysis section. The censoring time for these participants will be determined as the interval from baseline to their last recorded general practitioner (GP) visit date prior to the study endpoint (December 31, 2018).

Comment 6 Line 133-140: The matching process could be better explained, particularly how greedy matching was implemented (e.g., caliper width used).

Response: We sincerely appreciate the reviewer’s insightful comment. The greedy matching algorithm, implemented through the SAS macro, utilizes a hierarchical digit-based approach to propensity score matching, which inherently defines caliper widths by adjusting precision levels. The specified caliper width is 0.1. We have added the corresponding content in the Methods section.

Action: “The greedy matching algorithm, implemented through the SAS macro, utilizes a hierarchical digit-based approach to propensity score matching, which inherently defines caliper widths by adjusting precision levels. The specified caliper width is 0.1.” (Page 8, line 164-167, in the clean copy of the revised manuscript)

Comment 7

Results

Line 197: Consider reporting absolute risk reduction (ARR) along with HRs to aid clinical interpretation.

Response: We appreciate the reviewer’s suggestion. Per the reviewer’s suggestion, we have reported absolute risk reduction (ARR) in the Results section.

Action: “Participants with knee arthroplasty had a 23% lower risk of all-cause mortality compared with those without knee arthroplasty (HR=0.77, 95% CI 0.65 to 0.90; ARR=1.71%) (Table 1; Fig 1A). We identified 366 deaths (40.6 per 1,000 person-years) in subjects with hip arthroplasty and 395 deaths in those without hip arthroplasty (43.9 per 1,000 person-years) during follow-up. The mean follow-up period was 5.4 years for the hip arthroplasty group and 5.4 years for the non-hip arthroplasty group. The rate of all-cause mortality among subjects with hip arthroplasty was lower, albeit non-statistically significant, than those without hip arthroplasty (HR=0.87, 95% CI 0.73 to 1.04; ARR=1.49%) (Table 1; Fig 1B).” (Page 10, line 219-227, in the clean copy of the revised manuscript)

“Participants with knee arthroplasty had a 14% lower, albeit non-statistically significant, risk of CVD than their counterparts (HR=0.86, 95% CI 0.73 to 1.01; ARR=0.09%) (Table 2; Fig 2A). During the follow-up period, 135 cases of CVD in the hip arthroplasty group and 146 CVD cases in the comparison group were identified. Incidence rates per 1,000 person-years for the hip arthroplasty and comparison cohorts were 17.6 and 19.6, respectively. The mean follow-up period was 5.2 years for the hip arthroplasty group and 5.1 years for the non-hip arthroplasty group. The risk of CVD, albeit non-statistically significant, was lower in subjects with hip arthroplasty than those without hip arthroplasty (HR=0.84, 95% CI 0.69 to 1.02; ARR=0.77%) (Table 2; Fig 2B).” (Page 12, line 249-258, in the clean copy of the revised manuscript)

“Compared with those without knee arthroplasty, the HR of VTE for subjects with knee arthroplasty was 1.63 (95% CI 1.23 to 2.17) and the corresponding ARR was 0.97% (Table 3; Fig 3A). Similar results were also observed when the association between hip arthroplasty and the risk of VTE was examined (7.98 vs. 3.88 per 1,000 person-years). The mean follow-up period was 5.3 years for the hip arthroplasty group and 5.2 years for the non-hip arthroplasty group. The corresponding HR was 2.19 (95% CI 1.54 to 3.11) and ARR was 2.14% (Table 3; Fig 3B).” (Page 14, line 279-285, in the clean copy of the revised manuscript)

Comment 8 Tables 1-3: Ensure consistency in presenting statistical significance (bold or asterisk for p < 0.05).

Response: We are thankful for the reviewer’s feedback and the statistically significant results in Tables 1-3 have been bolded for emphasis.

Action: We have revised Table 1-3. (Page 10-15, in the clean copy of the revised manuscript)

Comment 9 Table 1 footnotes: Ensure uniform formatting for hazard ratios (HR) and confidence intervals (CI).

Response: Many thanks for the reviewer’s comment. We have revised the footnotes in Table 1 to ensure uniform formatting.

Action: We have revised Table 1 footnotes. (Page 11, line 233-234, in the clean copy of the revised manuscript)

Comment 10

Discussion

Line 275: “Our findings also showed that either knee or hip arthroplasty decreased the risk of CVD and increased the risk of VTE...” → Consider specifying the magnitude of risk reduction/increase directly in the sentence.

Response: We acknowledge the reviewer’s suggestion with appreciation. We have added the corresponding content to the Discussion section according to your suggestion.

Action: “Our findings showed a substantially lower risk of all-cause mortality (23% for KA, 13% for HA) in patients with RA after knee or hip arthroplasty, indicating significant protective associations for both KA and HA regarding all-cause mortality. Our study revealed that both KA and HA demonstrated cardioprotective properties in patients with RA, exhibiting a trend toward reduced cardiovascular disease risk (14% for KA, 16% for HA). However, this potential cardiovascular benefit must be weighed against a substantial elevation in VTE risk, with postoperative analyses showing 63% and 119% increased incidence rates for KA and HA recipients respectively.” (Page 15-16, line 305-313, in the clean copy of the revised manuscript)

Comment 11 Line 279: The discussion on mechanisms is strong but should also consider alternative explanations (e.g., RA patients undergoing arthroplasty may have better baseline health or access to care).

Response: We are thankful for the reviewer’s input. Per the reviewer’s comment, we have added the alternative explanations in the Abstract section.

Action: “Third, patients with RA undergoing arthroplasty may inherently have better baseline health or access to healthcare. Fourth, arthroplasty may attenuate systemic inflammation, a key driver of CVD in RA. Chronic inflammation, marked by elevated cytokines (e.g., IL-6, TNF-α) and acute-phase reactants like CRP, accelerates endothelial dysfunction and plaque instability 27,60. By alleviating localized joint inflammation and reducing inflammatory burden, arthroplasty may lower inflammatory burden, indirectly improving cardiovascular outcomes. Fifth, perioperative multidisciplinary care optimizes cardiovascular risk factors. Preoperative optimization of cardiac function, intraoperative hemodynamic monitoring, and postoperative rehabilitation protocols may mitigate surgical stress and enhance long-term outcomes 61.” (Page 17, line 345-355, in the clean copy of the revised manuscript)

References:

27 England, B. R., Thiele, G. M., Anderson, D. R. & Mikuls, T. R. Increased cardiovascular risk in rheumatoid arthritis: mechanisms and implications. BMJ 361, k1036, doi:10.1136/bmj.k1036 (2018).

60 Blum, A. & Adawi, M. Rheumatoid arthritis (RA) and cardiovascular disease. Autoimmunity reviews 18, 679-690, doi:10.1016/j.autrev.2019.05.005 (2019).

61 Heckert, S. L. et al. Long-term mortality in treated-to-target RA and UA: results of the BeSt and IMPROVED cohort. Annals of the rheumatic diseases 83, 161-168, doi:10.1136/ard-2023-224814 (2024).

Comment 12 Line 312: The authors should acknowledge potential unmeasured confounders, such as biologic DMARD use and socioeconomic factors.

Response: We appreciate the reviewer’s suggestion. We have included a discussion of potential unmeasured confounders, such as biologic DMARD use in the Discussion section of the manuscript to acknowledge this limitation. The socioeconomic factors, particularly socioeconomic status (Townsend Deprivation Index), were systematically adjusted for as potential confounders. We recognize that residual confounding may persist despite these methodological precautions, and would be happy to incorporate any additional specific suggestions the reviewer might have for further strengthening this aspect of our analysis.

Action: “Third, the current data set does not contain information on the period of biological disease-modifying anti-rheumatic drugs (DMARDs) administration. Since studies have shown that biological DMARDs use was associated a decreased risk of mortality and cardiovascular events in patients with RA 64,65, without adjustment of biological DMARD could also introduce potential residual confounding.” (Page 19, line 385-390, in the clean copy of the revised manuscript)

References:

64 Rodriguez-Rodriguez, L. et al. Treatment in rheumatoid arthritis and mortality risk in clinical practice: the role of biologic agents. Clin Exp Rheumatol 34, 1026-1032 (2016).

65 Greenberg, J. D. et al. Tumour necrosis factor antagonist use and associated risk reduction of cardiovascular events among patients with rheumatoid arthritis. Ann Rheum Dis 70, 576-582 (2011).

Comment 13 Line 374: The statement “coordinated care involving medicine, surgery, anaesthesia, and nursing throughout the perioperative process...” is important. Consider referencing specific perioperative guidelines for RA patients.

Response: We are thankful for the reviewer’s comment. Per the reviewer’s comment, we have added the corresponding reference regarding specific perioperative guidelines for RA patients to the Discussion section.

Action: “Therefore, it is essential for patients with RA who require surgery to have coordinated care involving medicine, surgery, anaesthesia, and nursing throughout the perioperative process, which facilitates the creation of a comprehensive interdisciplinary care plan before the operation 11,69,84.” (Page 22, line 472-475, in the clean copy of the revised manuscript)

“REFERENCES

11 Goodman, S. M. et al. 2017 American College of Rheumatology/American Association of Hip and Knee Surgeons Guideline for the Perioperative Management of Antirheumatic Medication in Patients With Rheumatic

---

## [Decision Letter · Decision Letter 1]

12 Aug 2025

Dear Dr. Wang,

Thank you for submitting your manuscript to PLOS ONE. After careful consideration, we feel that it has merit but does not fully meet PLOS ONE’s publication criteria as it currently stands. Therefore, we invite you to submit a revised version of the manuscript that addresses the points raised during the review process.

Several reviewers (notably Reviewers 2 and 5) have made detailed and constructive suggestions regarding the structure and content of the Abstract, Introduction, and Methods. Reviewer 2 emphasized the need to include additional epidemiological data (prevalence and mortality related to cardiovascular disease and venous thromboembolism in RA patients undergoing arthroplasty) and to clarify how cardiovascular and VTE diagnoses were determined. Reviewer 5 raised similar points about clarifying definitions, coding, and the inclusion criteria, and also stressed the need to adjust the title to reflect only statistically supported findings. These recommendations are consistent and should be addressed in the revision.

Reviewer 4 raised specific concerns about data presentation, including inconsistencies in patient numbers between tables and unclear statements about procedures performed. These issues are important for the transparency and credibility of the results, and I recommend you address them directly.

In some cases, reviewer suggestions diverge. For example, Reviewer 2 requested substantial additional intraoperative and subgroup analyses (tourniquet time, medications, subgrouping by BMI, gender, OA grade), whereas Reviewer 5 focused more on clarifying methodology, inclusion criteria, and the interpretation of results without requiring extensive new data collection. Given the nature of your study, which is based on a pre-existing database, some of Reviewer 2’s requests for additional intraoperative variables may not be feasible. In this case, I recommend you clearly explain the limitations of the database and why certain variables could not be included, rather than attempting unrealistic additional analyses.

Overall, please focus your revision on:

Clarifying the title and aligning it with statistically supported results.Revising the Abstract for consistency and specificity, including appropriate key words.Expanding the Introduction to address epidemiology and the rationale for the study.Providing methodological clarity on case definitions, coding, inclusion/exclusion criteria, and database characteristics.Correcting data inconsistencies in tables and figures.Shortening and refining the Discussion, ensuring that conclusions are balanced and do not imply causality where only associations have been demonstrated.Explaining clearly in the manuscript any variables not available in the dataset and the implications for interpretation.

  Addressing these points will improve the clarity, transparency, and clinical relevance of your work. I look forward to receiving your revised manuscript.

We look forward to receiving your revised manuscript.

Kind regards,

Gennaro Pipino, Md

Academic Editor

PLOS ONE

Journal Requirements:

Additional Editor Comments:

Dear Author,

Thank you for your dedication and the considerable effort you have invested in improving your manuscript. The review process for this submission has been particularly thorough and, at times, complex. The reviewers have acknowledged the significant revisions you have undertaken and have expressed appreciation for the progress made since the initial submission.

However, in order to ensure that the manuscript fully meets the standards and expectations of the journal, I kindly request that you address a final set of points in the form of a minor revision. These refinements will help to further strengthen the clarity, rigor, and overall quality of your work, thereby maximizing its impact for our readership.

I look forward to receiving your revised manuscript.

Reviewers' comments:

Reviewer's Responses to Questions

**Comments to the Author**

Reviewer #2: All comments have been addressed

Reviewer #3: (No Response)

Reviewer #4: All comments have been addressed

Reviewer #5: (No Response)

2. Is the manuscript technically sound, and do the data support the conclusions?

Reviewer #2: Partly

Reviewer #3: No

Reviewer #4: Yes

Reviewer #5: (No Response)

3. Has the statistical analysis been performed appropriately and rigorously?

Reviewer #2: N/A

Reviewer #3: I Don't Know

Reviewer #4: Yes

Reviewer #5: (No Response)

4. Have the authors made all data underlying the findings in their manuscript fully available?

Reviewer #2: Yes

Reviewer #3: No

Reviewer #4: Yes

Reviewer #5: (No Response)

5. Is the manuscript presented in an intelligible fashion and written in standard English?

Reviewer #2: Yes

Reviewer #3: No

Reviewer #4: Yes

Reviewer #5: (No Response)

Reviewer #2: Abstract

Row 27-29: This phrase could be more specific according with the main goal.

Key words: The authors could add more terms

Introduction

It is lacking information(frequency and prevalence) about cardiovascular problems eith RA and arthroplasty

the main causes of mortality associated with arthroplasty were not written.

The association of VTE and RA were poorly written in this section.

Methods

How was determined the quality of data(metadata)?

How was determined the diagnosis for CVD and VTE?

Results

It is lacking information about arthroplasty (tourniquet time, surgical time). How the intraoperative factors could be correlated with CVD and VTE?

The correlation about medications used used during the surgery and CVD and VTE were not mentioned. Why not?

It is lacking subgroups analysis ( gender, grade of osteoarthritis, laterality, grade of BMI)

Discussion

The comparison with variables like ethnicity, social status,nutrition, age were not mentioned . It could be more explored.

THe clinical relevance(CVD and VTE) associated with cost could be mentioned in this section.

Comparison with arthroplasty registries studies could be written in this section.

Conclusion

It can shortened according with the main findings

Reviewer #3: I find the article long, the title long, the short and long title are not aligned. Its not in clear concise english and I don't know what the key take away points are.

Reviewer #4: I have some comments.

1. Page 100-102, hypothesized that knee and hip arthroplasty may be associated with lower risks of mortality, it is sudden.

2. Page 210-214, “with and without knee arthroplasty”, did the patients performed arthroplasty or not?

3. Table 1-3, the number of patients is different (n=4,774 for knee arthroplasty, n=3,362 for hip arthroplasty, n=4,350 for knee arthroplasty, n=2,390 for hip arthroplasty, n=4,574 for knee arthroplasty, n=3,174 for hip arthroplasty). Are they the same patients?

Reviewer #5: 1. GENERAL

- This study concludes that total hip and knee replacement RA patients have lower mortality and cardiovascular risk, but higher risk of VTE.

2. TITLE

- The title should not include cardiovascular risk, as these findings were not statistically significant

3. ABSTRACT

- The authors stated in the conclusion that “carefully selected patients with RA” may have the benefits found from the study. Since this is a population based study, the authors cannot tell if these patients were carefully selected or not.

- The abstract on the 1st page is not the same as the abstract that is on page 21 after the title page

- The key words should include hip arthroplasty and knee arthroplasty, as arthroplasty encompasses multiple body parts. I would also add VTE and CVD (written out) if possible

4. INTRODUCTION

- Line 78: I would consider clarifying this sentence to state: Based on the anatomic location of their RA, patients with RA may undergo various surgical options, such as tenosynovectomy, radiosynovectomy, arthroscopic surgery, osteotomy, joint fusion, procedure for soft-tissue release, small-joint implant arthroplasty, metatarsal-head excision arthroplasty and total joint replacement.

- Line 100 – Typically the hypothesis is listed after the aim of the investigation and should be moved to Line 111.

5. MATERIALS AND METHODS

- For the IQVIA Medical Research Database, does this include data from NHS that follows patients longitudinally? What about patients who have private insurance or see physicians privately?

- Why did the authors only include patients greater than 20 years old? There are younger patients with RA who undergo joint replacement. It would be even more interesting to see the mortality in these patients.

- The authors accessed the data on July 14, 2022 – yet the timeframe for RA diagnosis was only until 2018. Why was this the case?

- Line 137 – A 60% overlap is not very good. What does that mean for the accuracy of diagnosis in the database?

- What coding was used to determine hip and knee replacements? We commonly use CPT codes. Were these total hip and total knee replacements? Were partial knee replacements included? Were total hip replacements for fracture included? Patients who undergo total hip for fracture are very different than elective patients.

- Is there any access to imaging records (e.g. Ultrasound) to record DVT?

- For VTE in the patient population, the findings may be skewed because these are patients in the UK only. How can risk factors for thromboembolism (e.g. Factor V Leiden, etc) affect patient outcomes? Was preoperative anticoagulation controlled for in the analysis?

6. RESULTS

- How do you define the mean time for follow-up for patients who did not undergo hip/knee arthroplasty? What was the timepoint that was defined as the start time? Many people are diagnosed with RA at much younger ages, so the follow up of approximately 5.5 years seems short for the non-arthroplasty group.

- KA and HA are not commonly used terms – it is typically total knee arthroplasty (TKA) and total hip arthroplasty (THA)

7. DISCUSSION

- I would avoid words like “pioneering” – line 302

- The discussion is very long and I would recommend reducing the volume of words

- Limitations – should add that there are no radiographs, so there is no ability to assess severity of the joint prior to surgery.

8. CONCLUSION

- The conclusion is problematic. First, I would avoid writing terms like “first evidence”. Secondly, the authors are hinting that patients with RA should get a joint replacement to reduce mortality. I think that’s worrisome as some patients may ask for a joint replacement to improve longevity, when it’s not a causal effect and it’s just correlation. Third, how do we know that these patients were carefully selected? It is impossible to tell in these population studies.

**Do you want your identity to be public for this peer review?** For information about this choice, including consent withdrawal, please see our Privacy Policy

Reviewer #2: No

Reviewer #3: No

Reviewer #4: No

Reviewer #5: No

---

## [Author Response · Author response to Decision Letter 2]

25 Sep 2025

Editors’ Comments to the Authors:

Comment 1 After careful consideration, we feel that it has merit but does not fully meet PLOS ONE’s publication criteria as it currently stands. Therefore, we invite you to submit a revised version of the manuscript that addresses the points raised during the review process.

Response: We appreciate the editor’s comment. We provide a point-by-point response to the reviewers’ comments. We hope that our responses are satisfactory and that the changes we have made in the text reflect our responsiveness to the comments and suggestions.

Comment 2 Several reviewers (notably Reviewers 2 and 5) have made detailed and constructive suggestions regarding the structure and content of the Abstract, Introduction, and Methods. Reviewer 2 emphasized the need to include additional epidemiological data (prevalence and mortality related to cardiovascular disease and venous thromboembolism in RA patients undergoing arthroplasty) and to clarify how cardiovascular and VTE diagnoses were determined. Reviewer 5 raised similar points about clarifying definitions, coding, and the inclusion criteria, and also stressed the need to adjust the title to reflect only statistically supported findings. These recommendations are consistent and should be addressed in the revision.

Response: Many thanks for the editor’s and reviewers’ suggestion. In light of the valuable feedback from the reviewers, we have made revisions to the structure and content of the Abstract, Introduction, and Methods sections. We have included requested epidemiological data, clarified the definitions and methodologies for cardiovascular/VTE diagnoses and inclusion criteria, and adjusted the title to reflect only statistically supported findings.

Comment 3 Reviewer 4 raised specific concerns about data presentation, including inconsistencies in patient numbers between tables and unclear statements about procedures performed. These issues are important for the transparency and credibility of the results, and I recommend you address them directly.

Response: We sincerely thank Reviewer 4 for the insightful comments. We have conducted a thorough re-examination of the dataset and the corresponding descriptions in the manuscript. All patient numbers have been carefully cross-checked and harmonized between the text and the tables to ensure complete consistency. Furthermore, the descriptions of the Methods and Results have been revised to be more precise and unambiguous. We believe these revisions have significantly enhanced the transparency and accuracy of our data presentation, and we have detailed all changes within the revised manuscript.

Comment 4 In some cases, reviewer suggestions diverge. For example, Reviewer 2 requested substantial additional intraoperative and subgroup analyses (tourniquet time, medications, subgrouping by BMI, gender, OA grade), whereas Reviewer 5 focused more on clarifying methodology, inclusion criteria, and the interpretation of results without requiring extensive new data collection. Given the nature of your study, which is based on a pre-existing database, some of Reviewer 2’s requests for additional intraoperative variables may not be feasible. In this case, I recommend you clearly explain the limitations of the database and why certain variables could not be included, rather than attempting unrealistic additional analyses.

Response: We sincerely appreciate the editorial guidance on reconciling the reviewers' comments. In revising the manuscript, we will carefully address the points raised by Reviewer 5 regarding methodological clarification, inclusion criteria, and interpretation of results, as these align well with the scope and data availability of our study. For the more extensive analytical requests from Reviewer 2—such as additional intraoperative variables and subgroup analyses—we will transparently acknowledge the limitations of our pre-existing dataset, clarifying where certain variables (e.g., tourniquet time, specific medication data) were not systematically collected and are therefore unavailable for analysis. This approach allows us to strengthen the paper's validity while maintaining methodological realism, and we will incorporate these explanations into the Discussion section of the manuscript.

Comment 5

Overall, please focus your revision on:

Clarifying the title and aligning it with statistically supported results.

Response: We appreciate the editor’s comment. We have clarified the title and aligned it with statistically supported results.

Action: “Reduced mortality but elevated venous thromboembolism risk following knee and hip arthroplasty in patients with rheumatoid arthritis: a general population-based cohort study.” (Page 1, line 1-3, in the clean copy of the revised manuscript)

Comment 6 Revising the Abstract for consistency and specificity, including appropriate key words.

Response: Many thanks for the editor’s comment. The Abstract has been revisied for consistency and specificity, including appropriate key words.

Action: “Arthroplasty is indicated for patients with rheumatoid arthritis (RA) who experience significant joint damage, including bone erosions, cartilage degradation and joint deformities. However, studies on its associations with all-cause mortality, cardiovascular disease (CVD), and venous thromboembolism (VTE) among patients with RA are scarce.” (Page 2, line 27-31, in the clean copy of the revised manuscript)

“Key words: Rheumatoid arthritis; Hip arthroplasty; Knee arthroplasty; Mortality; Cardiovascular disease; Venous thromboembolism” (Page 3, line 62-63, in the clean copy of the revised manuscript)

Comment 7 Expanding the Introduction to address epidemiology and the rationale for the study.

Response: Many thanks for the editor’s comment. We have expanded the Introduction to address epidemiology and the rationale for the study.

Action: “It is associated with increased risks of cardiovascular disease (CVD, 48%, pooled risk ratio=1.48 (95% CI 1.36 to 1.62)) 2,3, venous thromboembolism (VTE, 100%, adjusted hazard ratio=2.0 (95% CI, 1.9-2.2)) 4,5, and mortality (43%, HR=1.43 (95% CI 1.28 to 1.59)) 6,7. Epidemiological data indicate that approximately 9% patients with RA are affected by CVD, corresponding with an incidence rate of 3.30 per 100 patient-years (95% CI 2.08–4.25) 8, while VTE occurs around 7.2% of these patients 9.” (Page 4, line 67-73, in the clean copy of the revised manuscript)

“Previous studies have reported that the rate of mortality after hip and knee arthroplasty surgery is up to 4.8%, with major causes including pulmonary embolism, myocardial infarction, stroke, surgical site infections, prosthetic joint infections, and complications related to anesthesia and bleeding 19-22.” (Page 5, line 92-95, in the clean copy of the revised manuscript)

“However, no study has compared VTE incidence in patients with RA undergoing joint arthroplasty with those without joint arthroplasty. The increased susceptibility to VTE in RA is thought to result from chronic systemic inflammation, endothelial dysfunction, reduced mobility, and the use of glucocorticoids or NSAIDs 29-31.” (Page 5, line 107-111, in the clean copy of the revised manuscript)

“All-cause mortality is one of the most important indicators for the net risk-benefit effect of any clinical treatment regimens 18. Previous studies have reported that the rate of mortality after hip and knee arthroplasty surgery is up to 4.8%, with major causes including pulmonary embolism, myocardial infarction, stroke, surgical site infections, prosthetic joint infections, and complications related to anesthesia and bleeding 19-22. However, for patients with RA, joint arthroplasty may significantly improve physical function, reduce pain, and enhance overall quality of life. By alleviating disability and decreasing long-term systemic inflammation, such interventions could potentially contribute to a reduction in all-cause mortality among this population over time 23,24. Therefore, the association of either knee or hip arthroplasty with all-cause mortality in patients with RA specifically remains understudied.” (Page 5, line 90-101, in the clean copy of the revised manuscript)

“Based on the anatomic location of their RA, patients with RA may undergo various surgical options, such as tenosynovectomy, radiosynovectomy, arthroscopic surgery, osteotomy, joint fusion, procedure for soft-tissue release, small-joint implant arthroplasty, metatarsal-head excision arthroplasty and total joint replacement 14.” (Page 4, line 83-87, in the clean copy of the revised manuscript)

“The aim of this investigation was to evaluate the relationship of knee arthroplasty or hip arthroplasty to all-cause mortality, risk of CVD and incident VTE among patients with RA. To address these knowledge gaps, we conducted propensity-score matched cohort studies among patients with RA to investigate the relationship of knee or hip arthroplasty with risks of all-cause mortality, CVD and VTE, respectively, while controlling for potential confounders (including age, sex, BMI, lifestyle factors, and other comorbidities). Additionally, we performed subgroup analyses (differential risks by age and sex) to explore potential residual confounding by indication. Based on these, we hypothesized that knee and hip arthroplasty may be associated with lower risks of mortality and CVD but an increased risk of VTE among patients with RA.” (Page 6, line 117-127, in the clean copy of the revised manuscript)

Comment 8 Providing methodological clarity on case definitions, coding, inclusion/exclusion criteria, and database characteristics.

Response: We thank the editor for this insightful feedback. In response to the request for greater methodological clarity, we have provided methodological clarity on case definitions, coding, inclusion/exclusion criteria, and database characteristics. These amendments are incorporated to ensure the study's reproducibility and to bolster the transparency and rigor of our methodological approach.

Action: Revised the Methods section and the Results section. (Page 6-16, line 129-320, in the clean copy of the revised manuscript)

Comment 9 Correcting data inconsistencies in tables and figures.

Response: Many thanks for the editor’s suggestions. The description in the Results section has been revised accordingly to mitigate potential ambiguities, specifically to clarify which patients underwent arthroplasty and which did not.

Action: Revised the Results section. (Page 10-16, line 227-320, in the clean copy of the revised manuscript)

Comment 10 Shortening and refining the Discussion, ensuring that conclusions are balanced and do not imply causality where only associations have been demonstrated.

Response: We appreciate the valuable feedback. In revising this section, we have carefully shortened and refined the text to highlight the key findings and their context within the existing literature. Furthermore, we have ensured that our conclusions are appropriately balanced by consistently using language that reflects the observational nature of our study, clarifying that the results demonstrate associations rather than implying causal relationships. We believe these revisions will significantly strengthen the manuscript's clarity and scholarly rigor.

Action: Revised the Discussion section and the Conclusion section. (Page 16-22, line 322-471, in the clean copy of the revised manuscript)

Comment 11 Explaining clearly in the manuscript any variables not available in the dataset and the implications for interpretation.

Response: We are thankful for the editor’s input. To uphold methodological rigor, we will transparently acknowledge the limitations of our dataset, particularly that the analysis could not incorporate certain variables due to their unsystematic record. This candor is crucial for interpreting the findings realistically and will be elaborated upon in the Discussion section to strengthen the study's overall validity.

Action: “Several limitations of this study are worth noting. First, population-based studies such as those performed in IMRD often lack information on disease activity measures and disease characteristics. Although our study was restricted among subjects with RA, we were unable to assess the stage or gravity of disease between patients with and without knee or hip arthroplasty due to the constraints of the database used in this study. It is possible that patients with more severe disease have increased all-cause mortality and risk of CVD. Furthermore, due to the limitations of the IMRD database, we were unable to include data on the clinical pathways before arthroplasty, radiographs for the assessment of joint injury severity, VTE prophylaxis, imaging for DVT, preoperative cardiac evaluation and anticoagulation controlled, surgical details (e.g., tourniquet time, surgical time, type of implants, cementation or non-cementation and use of elastic stocking), and postoperative complications such as anemia and blood transfusions, which limits the scope of our analysis. Second, lack of sufficient information on the precise cause of death from some subjects precluded us from investigating the association of knee or hip arthroplasty with the risk of the cause-specific death. Nevertheless, the overall all-cause mortality trends are critically important in their own right, as mortality represents the overall net health outcome of various benefits and risks associated with disease management 18. Third, the current data set does not contain information on the period of biological disease-modifying anti-rheumatic drugs (DMARDs) administration. Since studies have shown that biological DMARDs use was associated a decreased risk of mortality and cardiovascular events in patients with RA 69,70, without adjustment of biological DMARD could also introduce potential residual confounding. Fourthly, we are unable to distinguish between unilateral and bilateral cases, and therefore cannot further explore the distinct impacts of unilateral and bilateral procedures on the outcomes. Fifth, we did not assess short-term postoperative complications (e.g., infections), future studies should be specifically designed to systematically investigate these outcomes.” (Page 18-19, line 384-411, in the clean copy of the revised manuscript)

Comment 12 Addressing these points will improve the clarity, transparency, and clinical relevance of your work. I look forward to receiving your revised manuscript.

Response: We sincerely thank the Editor for their constructive feedback and for the opportunity to revise our manuscript. We have carefully considered all the points raised and have diligently addressed each one in the revised version. The changes made aim to significantly enhance the clarity of our methodology, improve the transparency of our data presentation, and more strongly articulate the clinical relevance of our findings. We believe the manuscript has been substantially improved as a result of this process and we look forward to the Editor's assessment of the revised work.

Reviewer #2:

Comment 1

Abstract

Row 27-29: This phrase could be more specific according with the main goal.

Response: We thank the reviewer for this insightful comment. We have revised the opening sentence of the Abstract to more explicitly align with the primary objectives of our study. The phrase now specifically mentions the three key outcomes—all-cause mortality, cardiovascular disease, and venous thromboembolism—which are the focus of our investigation. This change better introduces the research gap our study aims to address.

Action: “Arthroplasty is indicated for patients with rheumatoid arthritis (RA) who experience significant joint damage, including bone erosions, cartilage degradation and joint deformities. However, studies on its associations with all-cause mortality, cardiovascular disease (CVD), and venous thromboembolism (VTE) among patients with RA are scarce.” (Page 2, line 27-31, in the clean copy of the revised manuscript)

Comment 2 Key words: The authors could add more terms.

Response: Many thanks for the reviewer’s suggestion. We have added more terms in the Key words section.

Action: “Key w

---

## [Decision Letter · Decision Letter 2]

12 Oct 2025

Reduced mortality but elevated venous thromboembolism risk following knee and hip arthroplasty in patients with rheumatoid arthritis: a general population-based cohort study

PONE-D-25-03390R2

Dear Dr. Wang,

We’re pleased to inform you that your manuscript has been judged scientifically suitable for publication and will be formally accepted for publication once it meets all outstanding technical requirements.

Kind regards,

Gennaro Pipino, Md

Academic Editor

PLOS ONE

Additional Editor Comments (optional):

I consider the manuscript ready for publication.

Reviewers' comments:

Reviewer's Responses to Questions

**Comments to the Author**

Reviewer #2: All comments have been addressed

Reviewer #3: (No Response)

Reviewer #4: All comments have been addressed

Reviewer #5: All comments have been addressed

2. Is the manuscript technically sound, and do the data support the conclusions?

Reviewer #2: Partly

Reviewer #3: No

Reviewer #4: Yes

Reviewer #5: Yes

3. Has the statistical analysis been performed appropriately and rigorously?

Reviewer #2: N/A

Reviewer #3: I Don't Know

Reviewer #4: Yes

Reviewer #5: I Don't Know

4. Have the authors made all data underlying the findings in their manuscript fully available?

Reviewer #2: No

Reviewer #3: Yes

Reviewer #4: Yes

Reviewer #5: Yes

5. Is the manuscript presented in an intelligible fashion and written in standard English?

Reviewer #2: No

Reviewer #3: No

Reviewer #4: Yes

Reviewer #5: Yes

Reviewer #2: Title

It did not call attention for main findings. The terms "reduced"and "elevated"are generic. I suggested to be more specific.

Abstract

Row 38- What it means "propensity"? Was it calculated?It is suggested to explain.

Row 58=60; The last phrase coul de removed.

Introduction

In this section, it is lacking proper justification for the study. What are the common sources of mortality associated with RA? Nothing was written. It can be improved.

Row 68-72: It is not recommended to write HR and CI. It is redundant. It can be removed.

Row 73-75. It is lacking references for this information. This phrase can be removed.

Row 77: What it means "high"? It is suggested to write with numbers and percentages.

Row 79-81: Nothing was written about lower limb alignment, range of motion, gait and knee stability. Why not?

Row 81: What it means "elevate"? It is suggested to write in numbers or pecentages.

Row 81: and "complicantions"? It is suggested to describe the common complications associated with RA and total arthroplasties. It can be improved.

The authors did not describe the classifications associated with different grades of joint degradation in RA. I suggest the authors to specify the types of RA that could be benefited with total arthroplasties. It can be improved.

Row 84-86. The surgical treatment for the advanced RA could be focus on arthroplasty. It is suggested to imptove thi phrase.

Row 92-93: This percentage was high in comparison with similar studies. The authors did not mention early(30 day) X late mortality. I suggest to scrutinize with more detailed information.

Row 117-126: The aim of the study was not stated clearly. The primary and secondary end points were not well defined. It is suggested to improve this topic in this paragraph.

Methods

It is lacking information about grade of RA, coronal and sagital deformity, flexion contracture, range of motion, previous treatment. tourniquet time and pressure, type of anesthesia, type of implants, level of constraint, amount of blood loss during the surgery, prophylaxis for VTE and rehabilitation after total joint. It is suggeted to add theses information for subgroup analyisis.

Results

This section can be shortened according with the relevant findings.

The variable age, why the authors chosen to share the subgroups 70 years old?

The complications related with readmissions, prolonged time during hospitalization were not written. Why not?

Discussion

There are few comparisons with early vs late mortality. It can be improved.

Preoperative information like medications for CVD, previous procedures and HB levels could be more explored in this section.

Surgical time, period of hopsitalization and readmission rate were not mentioned in this section. Why not?

The clinical relevance of the study remians unclear. It can be improved.

Conclusion'

It can be shoterned according with the goal and end points.

Reviewer #3: The title has been corrected to match statistically significant findings (now excludes CVD risk)

Abstract is aligned with study goals, includes appropriate key words, and avoids overstatement.

Introduction expanded with epidemiology and rationale, supported by strong references.

Methods clarified: case definitions, coding, inclusion/exclusion criteria, and database description are all well defined

Data presentation: inconsistencies across tables/figures have been addressed.

Discussion shortened, avoids causal language, and contextualizes results within RA, arthroplasty registries, cost implications, and disparities.

Limitations are transparently stated (lack of surgical details, DMARDs, unilateral vs bilateral procedures, etc.).

Ethics, funding, and competing interest statements are complete and PLOS-compliant.

Data availability statement is provided (“all relevant data are within the manuscript and supporting information

Minor issues that may still need tightening before final acceptance:

Abstract duplication – Reviewer 5 noted two abstracts (short one on page 1 vs full one later). Ensure only the correct full version is included in the final submission.

Clarity of patient cohorts – Though improved, Tables still use different denominators for outcomes (mortality, CVD, VTE). Make absolutely clear in table legends why numbers differ.

Language polish – Some sentences remain long/complex; a light copyedit for concise English may improve readability (a recurring reviewer concern).

Figures/Tables formatting – Ensure they meet PLOS style (self-contained, consistent n values, full footnotes).

Reviewer #4: Thank you for the opportunity to review the manuscript. The paper is very good. I have no comments.

Reviewer #5: The authors have addressed the points raised by the reviewers. They have made good point by point changes.

**Do you want your identity to be public for this peer review?** For information about this choice, including consent withdrawal, please see our Privacy Policy

Reviewer #2: No

Reviewer #3: **Yes: ** Nabeela Adam

Reviewer #4: No

Reviewer #5: No

---

## [Editor Report · Acceptance letter]

PONE-D-25-03390R2

PLOS ONE

Dear Dr. Wang,

I'm pleased to inform you that your manuscript has been deemed suitable for publication in PLOS ONE. Congratulations! Your manuscript is now being handed over to our production team.

Kind regards,

on behalf of

Professor Gennaro Pipino

Academic Editor

PLOS ONE